# Multimodal Comparison of Diabetic Neuropathy in Aged Streptozotocin-Treated Sprague–Dawley and Zucker Diabetic Fatty Rats

**DOI:** 10.3390/biomedicines11010020

**Published:** 2022-12-22

**Authors:** Annalisa Canta, Valentina A. Carozzi, Alessia Chiorazzi, Cristina Meregalli, Norberto Oggioni, Virginia Rodriguez-Menendez, Barbara Sala, Roberto Cosimo Melcangi, Silvia Giatti, Raffaella Lombardi, Roberto Bianchi, Paola Marmiroli, Guido Cavaletti

**Affiliations:** 1Experimental Neurology Unit, School of Medicine and Surgery, University of Milano-Bicocca, 20900 Monza, Italy; 2Department of Pharmacological and Biomolecular Sciences, University of Milan, 20100 Milano, Italy; 3Neuroalgology Unit, IRCCS Fondazione Istituto Neurologico “Carlo Besta”, 20100 Milano, Italy

**Keywords:** experimental diabetes, peripheral neuropathy, streptozotocin, Zucker rats

## Abstract

The development and progression of diabetic polyneuropathy (DPN) are due to multiple mechanisms. The creation of reliable animal models of DPN has been challenging and this issue has not yet been solved. However, despite some recognized differences from humans, most of the current knowledge on the pathogenesis of DPN relies on results achieved using rodent animal models. The simplest experimental DPN model reproduces type 1 diabetes, induced by massive chemical destruction of pancreatic beta cells with streptozotocin (STZ). Spontaneous/transgenic models of diabetes are less frequently used, mostly because they are less predictable in clinical course, more expensive, and require a variable time to achieve homogeneous metabolic conditions. Among them, Zucker diabetic fatty (ZDF) rats represent a typical type 2 diabetes model. Both STZ-induced and ZDF rats have been extensively used, but only very few studies have compared the long-term similarities and differences existing between these two models. Moreover, inconsistencies have been reported regarding several aspects of short-term in vivo studies using these models. In this study, we compared the long-term course of DPN in STZ-treated Sprague–Dawley and ZDF rats with a multimodal set of readout measures.

## 1. Introduction

Diabetes, one of the most frequent chronic diseases affecting nearly 500 million people worldwide [1], has entered the top 10 causes of death following a significant increase of 70% since 2000. Diabetes is also responsible for the largest rise in male deaths among the top 10, with an 80% increase since 2000 (https://www.who.int/news-room (accessed on 30 September 2022). These figures are particularly relevant in terms of general health policy, since untreated (or improperly treated) diabetes is invariably complicated by multi-organ damage. One of the most frequent and remarkable long-term complications of diabetes is represented by peripheral nerve damage, contributing to ulcers and subsequent amputations in the most severe cases [2,3,4]. Among the different types of nerve damage occurring in diabetes, symmetrical distal polyneuropathy (DPN) is the most frequent and troublesome, affecting more than 50% of diabetic patients [5]. Typical clinical features of DPN include sensory and motor impairment, frequently associated with neuropathic pain and autonomic dysfunction [6].

To date, suggested treatments from preclinical studies have failed to be effective in clinical trials, due to many concerns involving drug schedules, targeted mechanisms, inappropriate preclinical models, clinical study design, and disregarded risk factors [1,7].

The development and progression of DPN are due to multiple mechanisms, so the human disorder has a complex aetiology. The creation of reliable animal models able to reproduce the clinical condition remains challenging. In fact, a great number of animal models of DPN have been developed, but none mimics all the features of human disease [8]. For instance, the late neuropathological features of human peripheral neuropathy have no counterpart in rodent animal models [1].

However, despite some recognized differences with the human condition, most of the current knowledge on the pathogenesis of DPN relies on the results from rodent animal models [9]. The simplest and most widely used DPN model reproduces type 1 diabetes, established by streptozotocin (STZ)-induced massive chemical destruction of pancreatic beta cells. Spontaneous/transgenic models of diabetes are less frequently used, mostly because they are less predictable in clinical course, more expensive, and require a variable time to achieve homogeneous metabolic conditions [10]. Among them, Zucker diabetic fatty (ZDF) rats represent a typical type 2 diabetes model.

Both STZ-induced and ZDF rats have been used frequently, but very few studies have compared the long-term similarities and differences existing between these two models [11,12,13,14,15,16]. However, this information is crucial to allow a proper interpretation of the results and to propose a strong and reliable DPN pathogenesis hypothesis. Moreover, inconsistencies have been reported regarding several aspects of short-term in vivo studies using these models.

In 2014, consensus criteria about experimental design to ascertain the presence of diabetic neuropathy in rodent models were reached in a meeting of the Diabetic Neuropathy Study Group (Neurodiab) of the European Association for the Study of Diabetes (EASD). Among their recommendations, long duration of studies and the use of type 2 diabetes animal models were underlined, even if the STZ-treated rat was recognized as the most commonly employed model, with many advantages [17].

In this study, we compared the long-term course of DPN in STZ-treated Sprague–Dawley and ZDF rats with a multimodal set of readout measures, following most of the Neurodiab guidelines.

## 2. Materials and Methods

### 2.1. Animals

Male Sprague–Dawley (SD) rats (300–350 g at housing room arrival, Charles River Laboratories, Calco, Italy), as well as male fatty diabetic (fa/fa, ZDF) and lean non-diabetic (fa/+) Zucker rats (210–270 g, Charles River Laboratories, Calco, Italy), were used for the experiment. The care and husbandry of the animals were in conformity with institutional guidelines in compliance with national (D.L.vo n. 26, Gazzetta Ufficiale della Repubblica Italiana, March 3, 2014) and international laws and policies (2010/63/EU Directive; Guide for the Care and Use of Laboratory Animals, 8th ed., U.S. National Research Council, 2011). The experimental plan was preliminarily examined and approved by the ethics committee of the University of Milano-Bicocca (approval numbers 0023088 and 0032038). Animals were housed one per cage in a limited-access animal facility under set environmental conditions (55 ± 10% relative humidity and 22 ± 2 °C room temperature) and artificial 12 h light/dark cycles (07:00 a.m.–07:00 p.m.). The general condition of the animals was assessed daily and body weight was recorded once a week for the study period.

### 2.2. Diabetes Induction and General Assessment of Animal Conditions

In SD rats, diabetes was induced after overnight fasting using a single intraperitoneal injection of STZ (60 mg/kg of body weight, dissolved in sodium citrate buffer, pH 4.5, Sigma-Aldrich, Milan, Italy) [8]. Two days after STZ injection, urine glucose level was evaluated and only SD-STZ rats with levels > 270 mg/dL were included in the study. Untreated SD rats (SD-CTRL) were used as controls (n = 8/group).

ZDF rats were monitored weekly to detect the earliest changes in their glycaemic level, and it was planned to enrol them for the study when it reached a level of at least 200 mg/dL to achieve a group number = 10. The diabetic condition was evaluated by measurement of non-fasting tail vein blood glucose level using an Accu-Check Aviva blood glucose meter (Roche Diagnostic S.p.A., Mannheim, Germany). The ZDF group reached the target for being considered diabetic at the age of 8 weeks, as expected [1].

All rats had access to food and water ad libitum during the experimental period. Daily water and food intake were assessed by weekly weighing food and water for a period of observation of 48h at each time point. Accordingly, glycaemic level and animal weight were evaluated weekly (see flow chart in Figure 1).

### 2.3. Tail Nerve Conduction Velocity (NCV) Measurement

NCV was evaluated before and after 8 weeks from STZ injection in SD rats and at the age of 8 weeks in ZDF rats; subsequently NCV was measured after additional 8 and 16 weeks in both groups (Figure 1). NCV was determined in the tail of each animal as previously described in detail in several experimental paradigms [18,19,20].

All neurophysiological determinations were performed under standard conditions in a temperature-controlled room with animals kept under deep isoflurane anaesthesia and their body temperature was kept constant at 37 ± 0.5 °C using a warmed pad; vital signs and animal body temperature were monitored continuously.

### 2.4. Behavioural Test: Randall–Selitto Paw Withdrawal Test

Determinations were performed at the same time points as the NCV assessments, and 24 weeks after diabetes induction in SD rats or at the age of 24 weeks in ZDF rats (Figure 1). The mechanical nociceptive threshold was quantified in all animals for each experimental group using the Randal–Selitto paw withdrawal test with an analgesia meter (Ugo Basile, Comerio, Varese, Italy), which generates a linearly increasing mechanical force. Each animal was tested twice with a 30 min interval between the two evaluations; the test was performed first on the left and then on the right paw and the values were averaged. The results represented the maximum pressure (expressed in grams) tolerated by the animals.

### 2.5. Pathological and Morphometric Examination of Sciatic Nerve

At the end of the study period, the animals were sacrificed by CO2 inhalation followed by cervical dislocation. The left sciatic nerves were removed, fixed by immersion in 3% glutaraldehyde in 0.18 M PBS, postfixed in OsO4, epoxy resin embedded, and used for light microscope observation after toluidine blue staining, as previously described in detail [18,21,22]. Semithin (1 µm thick) sections were prepared from at least two tissue blocks for each animal. The sections were stained with toluidine blue and examined with a Nikon Coolscope light microscope (Nikon Instruments S.p.A, Campi Bisenzio, Italy).

Randomly selected toluidine blue-stained sections were used for morphometric assessment of the distribution of myelinated fibres and calculation of the ratio between the axonal/external fibre diameters (g-ratio). At least 300 fibres were measured in sciatic nerve sections belonging to 3 animals/groups. The same examiner performed measurements on coded specimens in order to ensure blindness as to the membership of the specimens to any diabetic or non-diabetic group.

### 2.6. Intraepidermal Nerve Fibre (IENF) Density

Peripheral nerve damage was assessed by IENF density quantification in the hind paw footpad skin [23]. Briefly, 3 mm round skin punch biopsies were collected from hind paws 16 and 32 weeks after diabetes induction in SD rats and at the age of 16 and 32 weeks in ZDF rats. Samples were immediately fixed in 2% paraformaldehyde–lysine-sodium periodate for 24 h at 4 °C, cryoprotected overnight, and serially cut with a cryostat to obtain 20 μm sections. Three sections from each footpad were randomly selected and immunostained with rabbit polyclonal antiprotein gene product 9.5 (PGP 9.5; Biogenesis, Poole, UK) using a free-floating protocol. One observer blinded to the healthy or neuropathic status of the rats independently counted the total number of PGP 9.5-positive intra-epidermal nerve fibres in each section under a light microscope at high magnification with the assistance of a microscope-mounted video camera. Individual fibres were counted as they crossed the dermal–epidermal junction and secondary branching within the epidermis was excluded. The length of the epidermis was measured using a computerized system (Microscience Inc., Seattle, WA, USA) and the linear density of the IENFs (IENF/mm) was obtained.

### 2.7. Na^+^/K^+^ ATPase Activity

Tibial stumps collected at sacrifice were dissected out, unsheathed, and homogenized in chilled solution containing 0.25 M sucrose, 1.25 mM EGTA, and 10 mM Tris, pH 7.5, at 1:20 (*w*/*v*) in a glass–glass Potter-Elvehjem homogenizer (DISA, Italy), and stored at −80 °C for ATPase determinations. Na^+^/K^+^ ATPase activity was determined spectrophotometrically as previously described [19]. Protein content in homogenates was analysed using Lowry’s method with bovine serum albumin as standard.

### 2.8. Myelin Proteins RNA Assay on Sciatic Nerve

RNA was prepared from sciatic nerves obtained at sacrifice using the Nucleospin RNA II kit (Macherey-Nagel, Cornaredo, Italy). RNA was analysed using a TaqMan qRT-PCR (quantitative real-time) instrument (CFX384 real-time system, Bio-Rad Laboratories, Segrate, Italy) using the iScriptTM one-step RT-PCR kit for probes (Bio-Rad Laboratories, Segrate, Italy). Samples were run in 384-well formats in triplicate as multiplexer reactions with a normalizing internal control (36B4 for P0 and PMP22, 18s rRNA for MBP). Probe and primer sequences were purchased from Eurofins MWG-Operon (Ebersberg, Germany) (P0 and PMP22) and from Life Technologies (Monza, Italy) (MBP).

### 2.9. Statistical Analysis

Differences observed in each investigated parameter in the 2 sets of animals were analysed by Student’s unpaired t-test (significance level set at *p* < 0.05) using GraphPad Prism v4 (GraphPad Software, La Jolla, CA, USA).

## 3. Results

### 3.1. General Assessment

Immediately after STZ injection, all SD-STZ rats developed a marked hyperglycaemia corresponding to values above 500 mg/dL in most animals detected 2 days after induction, which remained at a constant level throughout the entire period of observation (Figure 2A). Glycaemia in ZDF rats progressively and significantly increased in comparison with lean controls and at the age of 8 weeks it was above 200 mg/dL. The increase in glycaemia tended to reach a plateau, with most animals lying in the range of 400–500 mg/dL at the age of 16 weeks as well as at the subsequent observations (Figure 2A). Water intake and food consumption were consistent with the course of glycaemia in the two models (Figure 2B,C). These metabolic changes were initially paralleled by body weight changes. However, while SD-STZ rats’ weight was constantly lower than that observed in the control SD rats, ZDF rats had an early phase where they were heavier than lean controls, followed by a late phase starting at 16 weeks of age in which they progressively levelled their weight and eventually became significantly lighter than the lean control animals (Figure 2D).

### 3.2. Tail NCV Measurement

Tail NCV studies demonstrated a significant reduction in SD-STZ in comparison with untreated SD rats at the determination performed 16 weeks after diabetes induction (Figure 3A), and this result was also evident in 16-week-old ZDF animals (*p* < 0.001) (Figure 3A). When the comparison was repeated after 32 weeks, aged SD and lean Zucker rats showed an increase in NCV due to the expected complete maturation of the peripheral nervous system, while the NCV in SD-STZ and ZDF rats remained substantially unchanged and the difference vs. their controls was significant (*p* < 0.001).

### 3.3. Mechanical Threshold

Analysis of the mechanical threshold demonstrated a significant reduction indicating the development of mechanical hyperalgesia in SD-STZ rats examined 8 weeks after diabetes induction and in 8-week-old ZDF rats (*p* < 0.001 vs. respective controls) (Figure 3B). However, while the mechanical threshold tended to reach a plateau after 16 weeks of diabetes induction in all SD animals and in Zucker lean rats, ZDF rats progressively developed an increasingly severe mechanical hyperalgesia as they reached 24 and 32 weeks of age.

### 3.4. Pathological and Morphometric Examination of Sciatic Nerve

At the pathological level (Figure 4A,B), sciatic nerves obtained from SD-STZ rats 32 weeks after the onset of diabetes showed mild myelin irregularities and the density of myelinated fibres appeared normal, although the incidence of smaller myelinated fibres was increased (Figure 4C).

The pathological changes in 32-week-old ZDF rats were more severe and they involved both the myelin sheath and, to a lesser extent, the axon of myelinated fibres. Moreover, mild endoneural oedema was evident, leading to an apparent reduction in the density of myelinated fibres in comparison with lean controls (Figure 4D,E). The histogram of the distribution of myelinated fibres also confirmed in these animals that a shift toward smaller diameter was present in ZDF vs. lean control rats, although to a lesser extent if compared with SD-STZ rats (Figure 4F).

In both cases, calculation of the g-ratio confirmed that myelin thickness was significantly reduced in diabetic rats vs. their non-diabetic controls (SD-CTRL rats = 0.647 +/− 0.07 vs. SD-STZ rats 0.687 +/− 0.06, *p* < 0.01; lean controls 0.669 +/− 0.05 vs ZDF 0.696 +/− 0.06, *p* < 0.001).

### 3.5. IENF Density

IENF density was significantly reduced in the SD-STZ rats after 16 and 32 weeks from diabetes induction (−33%, *p* < 0.01 and −32%, *p* < 0.001, respectively, vs. SD-CTRL rats). Although the IENF density was not yet different in ZDF vs. lean controls at the age of 16 weeks, a similar reduction (−28%, *p* < 0.001 vs. lean controls) was evident in 32-week-old ZDF rats (Figure 5A).

### 3.6. Na^+^/K^+^ ATPase Activity

Sciatic nerve Na^+^/K^+^ ATPase activity was significantly reduced after 32 weeks from diabetes induction in SD-STZ rats (−39%, *p* < 0.001 vs. SD-CTRL rats) and, to a slightly lesser extent, also in 32-week-old ZDF rats (−22%, *p* < 0.01 vs. lean controls) (Figure 5B).

### 3.7. Myelin Protein RNA Assay

Gene expression of three important myelin proteins: P0, PMPP22, and MBP, were evaluated at the end of the study in the sciatic nerves of all animal groups. P0, PMP22, and MBP mRNA levels in the sciatic nerves of both SD-STZ and ZDF animals were not significantly changed vs. the respective controls (data available as Appendix A at https://board.unimib.it/research-data/ (available since 1 January 2023).

## 4. Discussion

The involvement of the peripheral nervous system in diabetic patients is frequent and potentially severe [4,6]. Therefore, a deeper knowledge of its pathogenesis is a major goal of preclinical research. Most of the available data on DPN has been obtained from animal models where marked hyperglycaemia is spontaneous or induced. These models are the most effective attempts at reproducing type 1 and type 2 diabetes, even if they scarcely replicate all the features of human DPN, contributing to translational failure. In fact, so far treatments based on preclinical study results have not been successful in clinical trials [8]. To improve the reliability of rodent DPN models for the elaboration of therapeutic strategies and enable collaboration among researchers, in 2014 the Diabetic Neuropathy Study Group (Neurodiab) of EASD developed a set of recommendations for the study and characterization of these models.

A recent review evaluating the application of the Neurodiab recommendations in more than 100 studies on peripheral neuropathy in type 2 diabetes mellitus rat models found their application in a limited number of articles. Frequently, not all of the three key DPN endpoints (behavioural test, neurophysiology, measure of nerve structure) have been implemented, and in many cases the observation period is limited (i.e., frequently just a few weeks after the onset of hyperglycaemia) [24]. However, complete knowledge of the long-term course of the disease is mandatory for reliably interpreting the results of studies designed to investigate the basic mechanisms of peripheral nervous system damage in diabetes or to test protective agents. In this study, we compared the features of DPN in vivo using the typical model of type 1 diabetes induced by STZ injection in SD rats and the ZDF rat model of type 2 diabetes, where homozygous missense mutation causes a non-functional leptin receptor. This kind of systematic, extensive comparison has not been performed so far (despite the fact that they are the two most frequent rat models of DPN [10,11,12,13,14,15,16,25,26,27,28]), even if they fail to reproduce strong demyelination and endoneural vessel wall thickening, two of the key features in human DPN [29].

Although in type 1 and type 2 diabetes the occurrence of DPN is likely due to different mechanisms [7], rodent models of both types of DPN show several similarities. Our study aimed to highlight the different features of DPN in the two models.

The course of the disease and the general conditions of the animals observed in our experiment are consistent with the few long-term results previously described [11,12,13,14,16,30], and they confirmed that despite the fact that very high levels of glycaemia were eventually achieved in both strains, differences exist between the SD-STZ and ZDF models.

The remarkable differences in the time course and extent of DPN in the SD-STZ vs. ZDF models parallel the course of hyperglycaemia (i.e., with a sudden onset in SD-STZ rats and progressive in ZDF animals). Overall, the serial assessments demonstrate that SD-STZ and ZDF type 2 diabetic rats show a similar impairment of NCV (reflecting large-calibre fibre damage), but ZDF rats have a more gradual reduction of IENF density (reflecting small-calibre fibre damage) as well as of the development of mechanical hyperalgesia than STZ-induced type 1 diabetes model. At the pathological level, despite a more gradual onset of DPN, the involvement of the nerve fibres was more evident in ZDF rats at the end of the observation period in comparison with SD-STZ rats, with the occurrence of endoneural oedema and more marked nerve fibre degeneration. The morphometric analysis confirmed a shift toward the smaller size of the myelinated fibres and reduction of myelin thickness in both strains as assessed by g-ratio calculation.

Most of the combined in vivo and post-mortem assessments here reported have not yet been described in detail in a prospective, parallel study in aged, diabetic rats, and comparison with the available literature data is therefore difficult [1,8,10,24]. However, partial results obtained in different studies on somatic function are consistent with our data and marked differences have been evidenced between the two models when the autonomic nervous system was investigated [31].

To achieve a better understanding of the possible implication of the use of the STZ and ZDF models, it must be considered that it is very likely that the different phenotypes and course of DPN evidenced in the two experimental models reflect different underlying pathogenic mechanisms, as extensively revised by Al-awar et al. [32] and by Yorek [7,8], and that is also the case for human DPN [33,34]. Metabolic mechanisms of hyperglycaemia with involvement of many pathways (glycation, protein kinase C, polyol, and others) have been widely studied in the pathogenesis of DPN. However, glycaemic control is less effective in reducing DPN in patients affected by type 2 diabetes compared to type 1 [34]. Thus, despite marked hyperglycaemia being a common feature, among the differences existing between diabetes type 1 and type 2 what seems to be particularly relevant is the role of lipids and, more broadly, of the “metabolic syndrome” typical of the latter, here reproduced using ZDF animals. In fact, hypertriglyceridaemia has been independently associated with sensory peripheral neuropathy, confirmed at the pathological level in nerve and skin biopsies [35]. Preclinical studies have supported the independent role of hyperlipidaemia in peripheral nerve function impairment, particularly regarding mitochondrial activity and axonal trafficking [36,37,38]. In fact, dyslipidaemia can induce changes in mitochondrial shape and activity. Moreover, while hyperglycaemia has no effect on their trafficking in sensory neurons, increased concentration of fatty acids impairs mitochondrial bioenergetics and trafficking along nerve axons. Not only sensory neurons but also Schwann cells are targeted by dyslipidaemia, showing biochemical changes in lipid metabolism that might be relevant to the onset and development of DPN.

Other potentially relevant events such as c-peptide changes are known to differ in animal models [39] as well as in type 1 vs. type 2 diabetes [40]. These considerations might have a very relevant impact on the interpretation, for instance, of the results of neuroprotective preclinical studies before their translation to the clinical setting.

The lack of biochemical investigations in our study is a limitation that does not permit making a correlation with the extensive phenotypic characterization here reported.

In conclusion, our extensive study allowed us to systematically compare a wide set of in vivo outcome measures in long-term diabetic rats, thus demonstrating significant differences during experimental DPN. These differences are potentially relevant in the interpretation of experimental studies based on these models and should be carefully considered when experimental studies are designed to investigate the pathogenesis of DPN or to test the effectiveness of neuroprotective treatments.

## Figures and Tables

**Figure 1 biomedicines-11-00020-f001:**
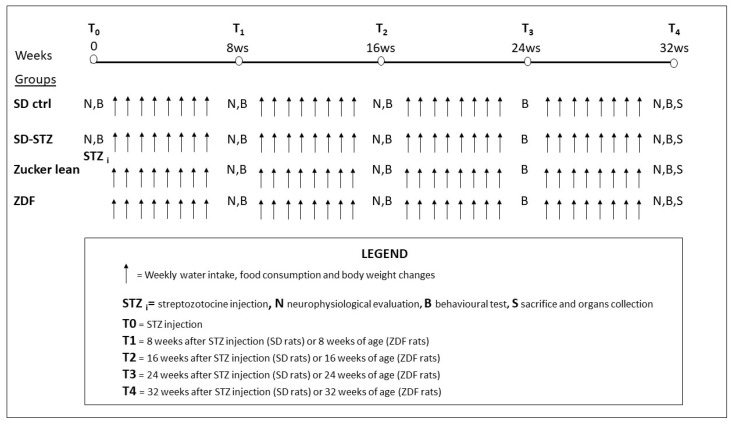
Flow chart of the study with the time points of each assessment.

**Figure 2 biomedicines-11-00020-f002:**
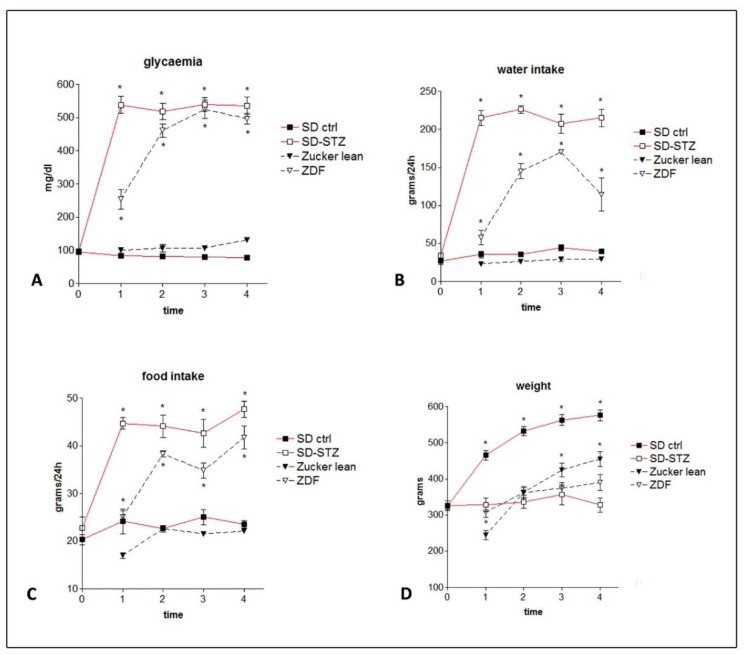
Weekly changes in glycaemia (**A**), food consumption (**B**), water intake (**C**), and body weight (**D**); SD-STZ and ZDF diabetic rats are compared with their non-diabetic controls at the different time points as follows: 0 = at the moment of STZ injection; 1 = 8 weeks after STZ injection (SD-STZ rats) or 8 weeks of age (ZDF rats); 2 = 16 weeks after STZ injection (SD-STZ rats) or 16 weeks of age (ZDF rats); 3 = 24 weeks after STZ injection (SD-STZ rats) or 24 weeks of age (ZDF rats); 4 = 32 weeks after STZ injection (SD-STZ rats) or 32 weeks of age (ZDF rats). Values represent mean +/− SEM (*) = *p* < 0.001.

**Figure 3 biomedicines-11-00020-f003:**
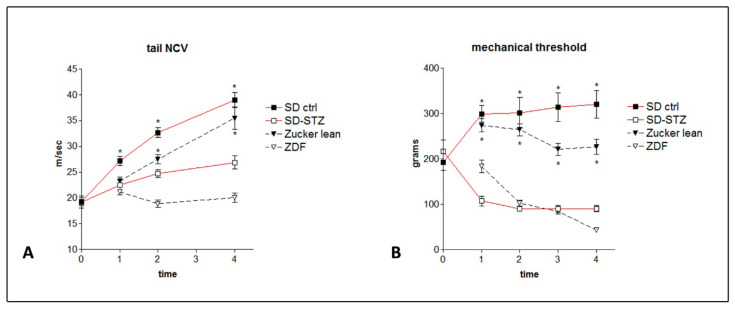
Changes in NCV (**A**) and mechanical threshold (**B**) over the period of observation; SD-STZ and ZDF diabetic rats are compared with their non-diabetic controls at the different time points as follows: 0 = at the moment of STZ injection; 1 = 8 weeks after STZ injection (SD-STZ rats) or 8 weeks of age (ZDF rats); 2 = 16 weeks after STZ injection (SD-STZ rats) or 16 weeks of age (ZDF rats); 3 = 24 weeks after STZ injection (SD-STZ rats) or 24 weeks of age (ZDF rats); 4 = 32 weeks after STZ injection (SD-STZ rats) or 32 weeks of age (ZDF rats). Values represent mean +/− SEM (*) = *p* < 0.001.

**Figure 4 biomedicines-11-00020-f004:**
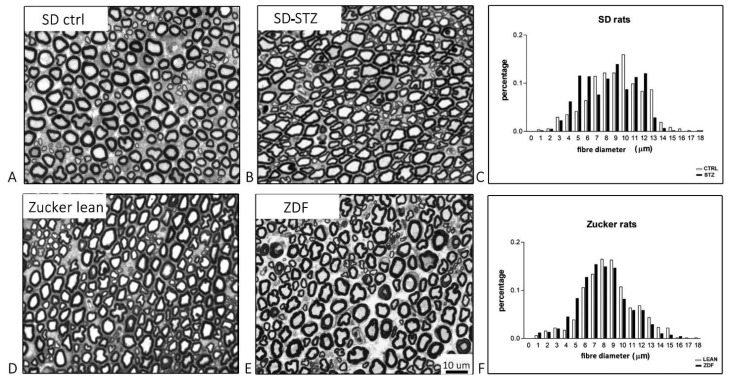
Representative light micrographs obtained from SD (**A**,**B**) and ZDF (**D**,**E**) rats with histograms of the distribution of myelinated fibres obtained in randomly selected sections (**C**,**F**). Morphometric analysis shows that diabetic rats (black bars) have an increased incidence of fibres of smaller diameter in comparison with the corresponding non-diabetic controls (white bars).

**Figure 5 biomedicines-11-00020-f005:**
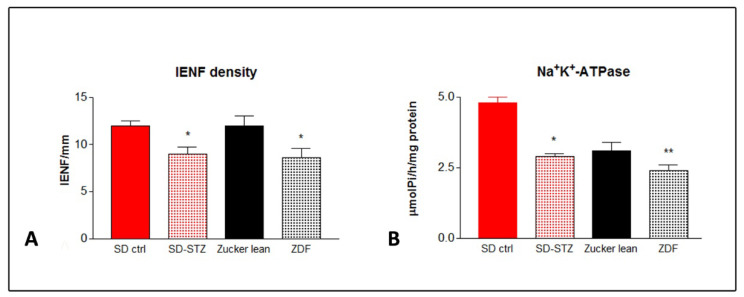
Changes in intraepidermal nerve fibre density (IENF) (**A**) and in Na+/K+ ATPase activity (**B**) at the end of the experiment in SD and ZDF rats. Values represent mean +/− SEM (*) = *p* < 0.001, (**) = *p* < 0.01.

## Data Availability

Data supporting the reported results can be found at https://board.unimib.it/research-data/ (available since 1 January 2023).

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
