# Peer review of "Multimodal Comparison of Diabetic Neuropathy in Aged Streptozotocin-Treated Sprague–Dawley and Zucker Diabetic Fatty Rats"

_biomedicines, 2022, doi:10.3390/biomedicines11010020_

Round 1
Reviewer 1 Report
This manuscript by A Canta et al., compares the pathological aspects and model progression differences between the animal models of diabetic neuropathy (DN) induced by Streptozotocin in SD rats (type I) and in Zucker fatty rats (type II). Despite the latest advances in pre-clinical research, our knowledge in DN area is limited because of the lack of a proper animal model that can perfectly replicate the features of human DN. In this regard, the current manuscript aimed to summarize the differences in the progression of the two most commonly used animal models of DN, is worthy of investigation. However, the major limitation of this study is that it fails to outline molecular characterization data and cellular mechanisms underlying DN pathogenesis. Instead, they sum up some basic neurobehavioral, functional and morphometric assessment for the two models. Though, this may be in accordance with the guidelines mentioned by European association for the study of Diabetes (EASD) on phenotyping DN in animal models published in 2014, tremendous amount of data has been generated in the past decade with respect to the contribution of various molecular events in the pathogenesis of DN. Hence, the current scientific community would be interested to read and learn about these latest advancements in the field and their possible contribution to the progression of both STZ model and the Zucker fatty rat model. The following are some of my concerns that authors may consider revising.
1. The data corresponds to IENF density, Na+-K+ ATPase activity, gene expression of myelin proteins need to be shown in the manuscript. It’s surprising that authors have discussed these findings without presenting them.
2. The detailed molecular characterization of the DN pathogenesis needs to be done and compared between these two models. For instance, various parameters related to oxidative stress, neuroinflammation, mitochondrial dysfunction, bioenergetic depletion, endoplasmic stress can be assessed for their preferential involvement in any of these models or in both should be described.
3. The Randall selitto test employed by the authors to assess the sensitivity to mechanical stimulation is indicative of mechanical hyperalgesia and doesn’t indicate mechanical allodynia. Please correct this in the results section.
4. It is also advisable to conduct Von Frey based mechanical allodynia assessment to supplement the neurobehavioral data.
5. Since there is also a role played by the lipids in the pathogenesis of DN, especially in type II diabetes, their levels need to be estimated at various time points to see how their levels correlates with the disease progression.
6. The line graphs can be color coded to distinguish different groups and the baseline data for the Zucker lean and Zucker fatty rats can be presented (if available) so all the lines appear to start from the same point.
Author Response
This manuscript by A Canta et al., compares the pathological aspects and model progression differences between the animal models of diabetic neuropathy (DN) induced by Streptozotocin in SD rats (type I) and in Zucker fatty rats (type II). Despite the latest advances in pre-clinical research, our knowledge in DN area is limited because of the lack of a proper animal model that can perfectly replicate the features of human DN. In this regard, the current manuscript aimed to summarize the differences in the progression of the two most commonly used animal models of DN, is worthy of investigation. However, the major limitation of this study is that it fails to outline molecular characterization data and cellular mechanisms underlying DN pathogenesis. Instead, they sum up some basic neurobehavioral, functional and morphometric assessment for the two models. Though, this may be in accordance with the guidelines mentioned by European association for the study of Diabetes (EASD) on phenotyping DN in animal models published in 2014, tremendous amount of data has been generated in the past decade with respect to the contribution of various molecular events in the pathogenesis of DN. Hence, the current scientific community would be interested to read and learn about these latest advancements in the field and their possible contribution to the progression of both STZ model and the Zucker fatty rat model. The following are some of my concerns that authors may consider revising.
- The data corresponds to IENF density, Na+-K+ ATPase activity, gene expression of myelin proteins need to be shown in the manuscript. It’s surprising that authors have discussed these findings without presenting them.
R: in order to save space we omitted in the previous version of the manuscript the results of IENF density and Na+-K+ ATPase activity, but now we added the Figure 5 reporting these significant results obtained at the end of the study. Since the data regarding gene expression of myelin proteins were always not significant, we would propose to add them just as Supplementary material (this is now indicated in the revised manuscript)
- The detailed molecular characterization of the DN pathogenesis needs to be done and compared between these two models. For instance, various parameters related to oxidative stress, neuroinflammation, mitochondrial dysfunction, bioenergetic depletion, endoplasmic stress can be assessed for their preferential involvement in any of these models or in both should be described.
R: we totally agree this issue is of pivotal importance in the study of diabetes complications since still unsolved; despite our study was not aimed at this very complex investigation, we added in the revised manuscript a clear acknowledgment of this crucial aspect (lines 329-333) and we also mention updated references for the Readers (refs. # 32-34)
- The Randall selitto test employed by the authors to assess the sensitivity to mechanical stimulation is indicative of mechanical hyperalgesia and doesn’t indicate mechanical allodynia. Please correct this in the results section.
R: we do apologize for this mistake done in 2 out of the 3 occasions where the test was mentioned, now corrected in the revised manuscript (lines 219 and 316)
- It is also advisable to conduct Von Frey based mechanical allodynia assessment to supplement the neurobehavioral data.
R: this is definitely a valuable comment: unfortunately we did not perform the Von Frey assessment in this study, but surely it will be added to our assessment toolkit in future studies
- Since there is also a role played by the lipids in the pathogenesis of DN, especially in type II diabetes, their levels need to be estimated at various time points to see how their levels correlates with the disease progression.
R: as for Point 4, we will consider this additional assessment in future studies
- The line graphs can be color coded to distinguish different groups and the baseline data for the Zucker lean and Zucker fatty rats can be presented (if available) so all the lines appear to start from the same point.
R: unfortunately, we did not record the baseline data in the Zucker rats; however, to make the graphs more clear, in this revised manuscript different colors have been used to separate SD from Zucker rats in Figures 2-5

Reviewer 2 Report
The current experimental study is intriguing. The researchers tried to find similarities and differences in diabetic peripheral neuropathy using two experimental animal models. The authors write that Streptozotocin (STZ) rats represent a typical type 1 diabetes model due massive chemical destruction of pancreatic beta-cells and Zucker diabetic fatty (ZDF) rats represent a specific type 2 diabetes model.
It has been hypothesized from the literature that there are differences in the expression of diabetic peripheral neuropathy in patients with type 1 and type 2 DM. The critical difference is the presence of c-peptide in patients with type 2 DM, and it is known that the c-peptide could be detected even in patients with long-duration DM. Moreover, the c-peptide has been used in the treatment of Diabetic Neuropathy in some studies with promising results.
So, the critical question is about the levels of c-peptide in both experimental models in the current study.
Furthermore, I think the authors should add a paragraph on the role of c-peptide in Diabetic neuropathy.
The methodology is appropriate. The manuscript is clearly written, and the discussion/conclusions are acceptable.
Overall, data are of interest.
Author Response
The current experimental study is intriguing. The researchers tried to find similarities and differences in diabetic peripheral neuropathy using two experimental animal models. The authors write that Streptozotocin (STZ) rats represent a typical type 1 diabetes model due massive chemical destruction of pancreatic beta-cells and Zucker diabetic fatty (ZDF) rats represent a specific type 2 diabetes model.
It has been hypothesized from the literature that there are differences in the expression of diabetic peripheral neuropathy in patients with type 1 and type 2 DM. The critical difference is the presence of c-peptide in patients with type 2 DM, and it is known that the c-peptide could be detected even in patients with long-duration DM. Moreover, the c-peptide has been used in the treatment of Diabetic Neuropathy in some studies with promising results.
So, the critical question is about the levels of c-peptide in both experimental models in the current study.
Furthermore, I think the authors should add a paragraph on the role of c-peptide in Diabetic neuropathy.
R: this comment regarding c-peptide is of great interest and importance; although we did not measure c-peptide levels, its importance has now been highlighted in the revised manuscript (lines 333-337), also adding new references (#35, 36)
The methodology is appropriate. The manuscript is clearly written, and the discussion/conclusions are acceptable.
Overall, data are of interest.
Reviewer 3 Report
The manuscript by Canta et al. needs to be improved before being further considered for a publication in Biomedicines. I suggest extensive manuscript style and language editing and clarification of numerous ambiguous sentences, in particular. Current version of the manuscript is not sufficiently clear, despite high relevance of the presented study to the scientific community studying diabetic polyneuropathy. Some of the suggestions to be addressed:
1. Abstract (line 20): Please use the abbreviation DPN when mentioning diabetic polyneuropathy for the for the first time
2. Introduction: Please correct: line 21 "(...) and the creation of the animal model is challenging"- sentence transition is unclear
3. Introduction: Please correct: line 50-53- same as above
4. Introduction: Line 57- please replace "no one" with "none"
5. Introduction: Line 61- please replace "results achieved using" with "results from"
6. Introduction: Line 71- what do the authors mean by the "sound pathogenic hypothesis"?
7. Figures- the x and y axes are not clearly visible- please correct
8. Please change the labeling- Figure 1 should start from results, experimental plan should not be included into the figures
9. Words, e.g., in vivo- should be italicized
10. Please use blood glucose concentration units consistently throughout the text, e.g., mg/dL only
11. Line 96- please replace "fast" with "fasting" and correct the whole sentence to make it clear
12. Discussion- which of the two models described by the authors more closely resembles human DPN? Please discuss
Author Response
The manuscript by Canta et al. needs to be improved before being further considered for a publication in Biomedicines. I suggest extensive manuscript style and language editing and clarification of numerous ambiguous sentences, in particular. Current version of the manuscript is not sufficiently clear, despite high relevance of the presented study to the scientific community studying diabetic polyneuropathy. Some of the suggestions to be addressed:
R: the text has been extensively revised for clarity and language
- Abstract (line 20): Please use the abbreviation DPN when mentioning diabetic polyneuropathy for the the first time
- Introduction: Please correct: line 21 "(...) and the creation of the animal model is challenging"- sentence transition is unclear
- Introduction: Please correct: line 50-53- same as above
- Introduction: Line 57- please replace "no one" with "none"
- Introduction: Line 61- please replace "results achieved using" with "results from"
- Introduction: Line 71- what do the authors mean by the "sound pathogenic hypothesis"?
R: all the very proper remarks 1-6 (as well as a few others identified during the text revision) allowed us to improve the manuscript
- Figures- the x and y axes are not clearly visible- please corr in this revised version
R: the font size has been increased from 10 to 12 in figures 2-5
- Please change the labeling- Figure 1 should start from results, experimental plan should not be included into the figures
R: if not mandatory, we would prefer leaving figures’ labeling as in the original version
- Words, e.g., in vivo- should be italicized
R: in vivo and post-mortem have now been indicated in Italics
- Please use blood glucose concentration units consistently throughout the text, e.g., mg/dL only
R: Blood glucose concentration was already indicated only in mg/dl, while osmolarity was used for urine testing: however, now we agree that this might be misleading and unclear to the reader and we have indicated also this measurement in mg/dl (line 104) to make the text more consistent, as suggested
- Line 96- please replace "fast" with "fasting" and correct the whole sentence to make it clear
R: the text has been revised accordingly
- Discussion- which of the two models described by the authors more closely resembles human DPN? Please discuss
R: as mentioned in the text, both models mimic only partially type 1 or type 2 diabetes; we do not believe it is possible to really indicate by direct comparison which one best suits the two different diabetes types. Thus, the aim of our study was to outline differences between the two models in the long-term course of DPN to be considered in future experimental research.

Round 2
Reviewer 1 Report
In this revised manuscript, authors have not addressed all of my concerns.
In the event that authors can't perform the molecular characterization of the two models, they should try to summarize these possible mechanistic differences (based on the literature evidences) and potentially discuss this as a limitation of their study.
They said that they discussed these things in the discussion (lines 329-331), however, I couldn't find any changes in those lines. This should be definetely included in the manuscript.
Author Response
In this revised manuscript, authors have not addressed all of my concerns.
In the event that authors can't perform the molecular characterization of the two models, they should try to summarize these possible mechanistic differences (based on the literature evidences) and potentially discuss this as a limitation of their study.
They said that they discussed these things in the discussion (lines 329-331),however, I couldn't find any changes in those lines. This should be definetely included in the manuscript.
R. We agree that mechanistic investigation is a very important point and the lack of molecular characterization of the two models could be considered a limitation of the study, but this complex analysis was beyond the aim of our study. A sentence regarding this limitation in the study was added at line 354. Moreover, based on the literature data, in the revised manuscript we added a quite extensive part about pathogenetic mechanisms in the discussion (lines 333-355), with relative references.
Thank you very much for your reply and contribution to improve the manuscript
Reviewer 2 Report
The authors answered to my comments.
Author Response
The authors answered to my comments
R. Thank you very much for your reply and contribution to improve the manuscript
Reviewer 3 Report
The authors addressed all the concerns. The only change required- please replace "pathogenic hypothesis" (illogical) with "DPN pathogenesis hypothesis"
Author Response
The authors addressed all the concerns. The only change required- please replace"pathogenic hypothesis" (illogical) with "DPN pathogenesis hypothesis".
R. We apologize for this mistake, now corrected in the revised manuscript (line 73).
Thank you very much for your reply and contribution to improve the manuscript